# Long-Term Clinical and Multimodal Imaging Findings in Patients with Disseminated Mycobacterium Chimaera Infection

**DOI:** 10.3390/jcm10184178

**Published:** 2021-09-16

**Authors:** Sandrine Anne Zweifel, Maximilian Robert Justus Wiest, Mario Damiano Toro, Pascal Hasler, Peter Maloca, Barbara Hasse, Nina Khanna, Robert Rejdak

**Affiliations:** 1Department of Ophthalmology, University Hospital Zurich, 8091 Zurich, Switzerland; maximilian.wiest@usz.ch (M.R.J.W.); toro.mario@email.it (M.D.T.); 2Department of Ophthalmology, University of Zurich, 8091 Zurich, Switzerland; 3Department of General and Pediatric Ophthalmology, Medical University of Lublin, 20079 Lublin, Poland; robertrejdak@yahoo.com; 4Institute of Molecular and Clinical Ophthalmology Basel, 4023 Basel, Switzerland; pascal.hasler@usb.ch (P.H.); info@dr-maloca.ch (P.M.); 5Department of Ophthalmology, University Hospital Basel, 4023 Basel, Switzerland; 6Division of Infectious Diseases and Hospital Epidemiology, University Hospital Zurich, 8091 Zurich, Switzerland; barbara.hasse@usz.ch; 7Division of Infectious Diseases and Hospital Epidemiology, University Hospital Basel, 4023 Basel, Switzerland; nina.knanna@usb.ch

**Keywords:** *Mycobacterium chimaera*, multimodal imaging, multifocal choroiditis

## Abstract

Background: To analyze long-term ophthalmic clinical and multimodal imaging findings of disseminated *Mycobacterium (M.) chimaera* infection after cardiothoracic surgery among the Swiss Cohort. Methods: Systemic and multimodal ophthalmic imaging and clinical findings including rate of recurrence were reviewed and correlated to a previously proposed classification system of choroidal lesions and classification of ocular disease. Main Outcomes Measures: long-term clinical and multimodal ocular imaging findings of *M. chimaera*. Results: Twelve patients suffering from systemic infection from *M. chimaera* were included. Mean age at the first ophthalmic examination was 59 years (range from 48 to 66 years). Mean duration of the follow-up was 22.63 ± 17.8 months. All patients presented with bilateral chorioretinal lesions at baseline; 5 patients had additional signs, including optic disc swelling (2), choroidal neovascularization (1), retinal neovascularization (1) and cilioretinal vascular occlusion (1). Four recurrence events after discontinuation or adjustment of the antibiotic treatment were observed. Progressive choroiditis was seen in 5 patients under treatment, 4 of them deceased. Conclusions: Expertise from ophthalmologists is not only relevant but also critical for the assessment of the adverse drug effect of antimycobacterial treatment along with monitoring therapeutic response and identifying recurrences.

## 1. Introduction

*Mycobacterium (M.) chimaera* is one of the recently described [1], ubiquitous, non-tuberculous mycobacterium (NTM), belonging to the *Mycobacterium avium* complex [2]. Prior to the current global outbreak of disseminated *M. chimaera* among patients who underwent open-chest surgery with cardiopulmonary bypass (CPB), *M. chimaera* was known as an opportunistic human pathogen. This was known to generally cause lung infection in individuals with underlying lung disease and disseminated infections in severely immunocompromised patients [1]. In 2013, *M. chimaera* was identified as a possible contaminant agent causing prosthetic valve endocarditis, and its way of transmission was linked to heater-cooler devices (HCDs) used in CBP [3]. Since the detection of the route of transmission of this microorganism, the number of diagnosed *M. chimaera* cases is increasing [4,5]. As per the reports from September 2017, globally nearly 120 cases of the infection by *M. chimaera* have been reported after cardiothoracic surgery [6]. *M. chimaera* diagnosis is usually delayed, sometimes after several years of cardiac surgery. The delay in diagnosis could be attributed to many factors such as time disparity between surgery and onset of infection symptoms, sub-acute presentation of the infection, lack of appreciation for the disease, intermittent bacteremia, low percentage positivity in tissue culture and observation of a normal echocardiogram in the early stages of the infection [7]. In spite of the fact that detection methods based on culture are the gold standard for diagnosis, these methods are time consuming and require a special culture medium for growth of the bacteria [7]. Moreover, the crude mortality of the outbreak has been very high, close to 50%. Many patients undoubtedly die before the diagnosis [7].

Disseminated infections caused by *M. chimaera* have been reported after many types of open-chest surgeries for CPB and were reported to have ocular involvement [7]. The clinical and ocular findings related to the disseminated infection by *M. chimaera* were reported in 2017, which suggested choroidal manifestations as an important finding of the infection [8,9]. A high morbidity and mortality rate has been found to be associated with infection by *M. chimaera* in patients who underwent cardiopulmonary bypass surgery, with a fatality rate of 50% [6]. This indicates that ophthalmologists should therefore be aware of the ocular and systemic findings, and play an important role in the multidisciplinary diagnosis and treatment approaches [7]. Classification of the choroidal lesions into active or inactive lesions based on multimodal imaging is the deciding factor for treatment regimen in these patients [10]. Recently, the recommendations for the screening and follow-up of the ophthalmological examinations were published for the patients who are suspected for or have confirmed *M. chimaera* infection [10].

Nevertheless, in spite of the serious impact of this often-lethal disease, there are no data on long-term effects so far. Therefore, this study was designed to report the long-term ophthalmic clinical and multimodal imaging findings of disseminated *M. chimaera* infection after cardiothoracic surgery among the Swiss Cohort.

## 2. Materials and Methods

The study was approved by the institutional review board (Cantonal ethics committee of Zurich, BASEC-N° 2019-02043), and the tenets of the Declaration of Helsinki were followed. A written informed consent was obtained from each patient. Patients suffering from infection after cardiothoracic surgery were identified, using methods previously described by other Swiss research groups [4,5]. In brief, patients who underwent open-heart surgery were diagnosed for *M. chimaera* infection based on blood cultures, tissue cultures or 16SrRNA polymerase chain reaction (PCR). Only patients who underwent ophthalmological examinations at any point after diagnosis were included. In this study, patients examined at the University Hospital of Zurich and the University Hospital of Basel were included.

Data regarding fundus biomicroscopy, optical coherence tomography (OCT), enhanced depth imaging OCT (EDI-OCT), OCT angiography (OCTA), fluorescein and indocyanine green angiography (FA, ICGA), fundus autofluorescence (FAF), color fundus photography (FP) and ultra-wide field (UWF)FP were included. In addition, electronic health records were reviewed, extracting information about the antimicrobial therapy and evidence of accompanying systemic disease activity. The method of image acquisition and assessment of ocular findings regarding progressiveness and activity of the infection was previously described by Böni et al. [10]. After diagnosis of ocular involvement, ophthalmic investigations were usually performed every 2 to 4 months or more frequently, as per the indications.

The Best Corrected Visual Acuity (BCVA) was measured using the Snellen chart and was converted to ETDRS (Early Treatment Diabetic Retinopathy Study) letters. Each line on this chart represents a change of five letters. A change of at least ≥5 letters was considered significant [11].

The presence of only very few inactive (2–5/eye) choroidal lesions was designated as mild ocular abnormality. “Reactivation” was referred to as the recurrence of the disease activity after adaptation/stopping of therapy and “breakthrough” referred to a recurrence of disease activity under continuous therapy, as defined in a previous report [10].

Data at the baseline were registered, and all subsequent ophthalmic examinations were included. In addition, data on the systemic course of disease were collected.

## 3. Results

Ten (20 eyes) out of 14 patients with known disseminated *M. chimaera* infection were included in this study. Four patients were excluded because an ophthalmological examination could not be performed. Required patient number was maintained in coherence with the previous publications, allowing readers from the ophthalmological community and other specialties to correlate ophthalmic findings to other published systemic data [5,8,10]. Mean age of the recruited cohort was 59 (range from 48 to 66 years) at the ophthalmological baseline visit. Nine of ten patients were male (90%). Mean follow-up time was 29.59 (±27.35) months. The maximum follow-up period observed in a patient was 88 months (case 3). Four of ten patients (40%) died after a mean follow-up time of 18.33 (±15.6) months. One patient (case 2) died 6 days after the baseline visit due to the splenic rupture and disseminated mycobacterial infection. Visual acuity at the baseline ophthalmic visit was 74.95 (±19.72) ETDRS letters. At last follow-up, mean visual acuity was stable in 10 (50%) eyes, had improved in 7 (35%) eyes and decreased > 5 letters in 2 (14.3%) eyes (case 5). The decrease in visual acuity was related to the granulomas at the posterior pole and cataract.

All ten patients presented with bilateral chorioretinal lesions in the posterior pole and the periphery at baseline (Figure 1). Five of the 10 patients presented with progressive systemic *M. chimaera* disease, while 5 presented with quiescent systemic *M. chimaera* disease. Chorioretinal *M. chimaera* lesions were observed in all the patients, both in the posterior pole and the periphery.

All patients were undergoing antimycobacterial treatment with a 4- or 5-fold tuberculostatic therapy containing clarithromycin, ethambutol, rifabutin, and moxifloxacin ± amikacin at the baseline ophthalmic visit. The goal of an early redo surgery with replacement of all the cardiac foreign material was pursued, if possible.

During follow-up, treatment was discontinued in 5 (50%) patients after 17 (±11) months because of clinically stable disease after replacement of the infected material. As per the definition of a reactivation or a disease breakthrough [10], 5 (50%) out of ten patients developed a breakthrough under triple or quadruple therapy; 4 (80%) of the 5 patients with a breakthrough infection died after a mean period of 113 (±171) days. Two (20%) patients suffered from a relapse of *M. chimaera* disease immediately after modification or intermission of therapy.

During follow-up, two patients developed a neovascularization (NV) (case 3, left eye retinal NV after 86 months (Figure 2); case 4, type 2 macular NV right eye after 36 months, (Figure 3)) [12].

One patient presented himself with a cilioretinal vessel occlusion at the 18 months follow-up (case 12, right eye, at 18 months follow-up (Figure 4)).

No additional ocular complications related to the systemic manifestations of the disease were detected during the entire follow-up in any patients.

The number of chorioretinal lesions observed in our cohort ranged from solitary lesions in ultra-widefield (UWF) FA/ICGA (case 3, right eye, Figure 1) to widespread (>100 lesions) detected in UWF-scanning laser ophthalmoscopy (left eye, case 4, Figure 1). At the end of the follow-up, case 3 presented no new lesions. Case 4 developed multiple new lesions before his last follow-up. A detailed case presentation of ocular findings in many cases has not been previously described [10]. However, these details are presented in Figure 1 in the present study. Clinical and demographic characteristics of the patients are detailed in Table 1.

### 3.1. Case 3

A 72-year-old man who had undergone mitral valve reconstruction presented with disseminated *M. chimaera* infection 39 months after the surgery. A bilateral mild uveitis with 0.5+ anterior chamber cells and 0.5+ vitreous cells was diagnosed on ophthalmic examination 5 years post cardiac index surgery. Similar to the lesions observed in the case 4, only a few and subtle chorioretinal lesions (2–5 lesions per eye) were observed in this case (Figure 1). Normal visual acuity was observed (20/20). A biological prosthesis was used in place of a mitral annuloplasty ring as a replacement, and the patient was kept on tuberculostatic therapy for more than 1 year after the redo surgery. At 21 months, after initial ocular diagnosis, the patient developed new lesions in the superonasal quadrant of the right eye and had recurrent fever episodes. After re-initiation of an empirical antibiotic therapy with penicillin, daptomycin and gentamicin, the ocular findings stabilized. Antibiotic therapy with daptomycin and later minocycline was continued for more than 1 year. The patient is currently under post-treatment follow-up. Ocular visits till now have shown inactive atrophic chorioretinal lesions without the occurrence of new lesions 62 months after the diagnosis of ocular involvement. However, the patient developed a retinal neovascularization at 58 months after initial diagnosis.

### 3.2. Case 9

This is the case of a now 55-year-old male with disseminated *M. chimaera* infection who was diagnosed with chorioretinal lesions in the right eye. At first presentation, 33 months after the index mitral valve annuloplasty and 2 months after the second mitral valve annuloplasty, the slit lamp exam revealed a quiet anterior chamber in both the eyes. In the left eye, the vitreous showed some isolated cells but no active vitritis, and no chorioretinal lesions were noted. In the right eye, the patient presented with distinct white lesions nasally. At this point, the patient was under therapy with clarithromycin, rifabutin, ethambutol and moxifloxacin for 6 months. After 30 months of stable disease with normal positron emission tomography–computed tomography (PET-CT) scan, echocardiography, laboratory values and negative blood cultures under quadruple therapy, antibiotic treatment was halted. Two weeks later, the patient experienced fever and fatigue, but had no ocular symptoms. A routine ophthalmic examination 4 weeks after the end of the therapy showed multiple new chorioretinal lesions in both the eyes (over 20 each). After detection of new ocular lesions, therapy with clarithromycin, ethambutol, bedaquiline and rifabutin was reinitiated. In the following 5 months of follow-up, lesions were stabilized with no new detectable lesions. After 5 months, bedaquiline was stopped due to the side effects (mood instability). In the meantime, the patient had a cerebral infarction. Echocardiogram and PET-CT scan did not show any signs of endocarditis. Noticeably, the ophthalmic examination showed new lesions. With new chorioretinal lesions now being one of the main clinically active parameters in surveilling this patient, rifabutin was stopped and bedaquiline and amikacin were reintroduced to the therapeutic regime. During the follow-up, therapy was changed to ethambutol, clarithromycin and bedaquiline. Twenty-three months after reactivation, the patient showed stable findings under this triple regimen. A second re-surgery with mitral valve reconstruction was performed 19 months (72 months after index surgery) after reactivation. *M. chimaera* grew on the annuloplasty ring, thereby confirming the persistent colonization of *M. chimaera* after the first re-surgery. To date, 86 months after index surgery, the patient was treated with clarithromycin, ethambutol and bedaquiline and has shown no signs of active lesions in the last FA/ICGA examinations.

### 3.3. Case 11

A 55-year-old female showed up with recently diagnosed ocular involvement of the disseminated *M. chimaera* infection post cardiac valve replacement surgery. Ocular lesions were first discovered 2 months after the initial diagnosis. Disseminated lesions were also detected in the liver, lungs, spleen and brain. She presented with vitreous and anterior chamber inflammation in addition to the bilateral chorioretinitis. Quadruple therapy (clarithromycin, ethambutol, bedaquiline and rifabutin) was given for 4 months, then ethambutol was stopped. The patient showed continuous development of new chorioretinal lesions in both eyes for the first 4 months. In the latest follow-up, 6 months after the initial infection, ocular involvement stabilized under clarithromycin, bedaquiline and rifabutin.

## 4. Discussion

Although cardiac surgeries have ensured the survival of many patients, unexpected sequelae due to the systemic infection by *M. chimaera* have been reported, which could reverse the success. Because of the easy access to the vascular system through the eyes, a multidisciplinary approach involving ophthalmology diagnosis and monitoring takes a key position for controlling *M. chimaera* infection [7]. Typically, *M. chimaera* infected patients present with various non-specific symptoms depending on the involvement of the different organs. This is due to the disseminated granulomatous inflammatory manifestations including endocarditis, vascular graft infection, bacteremia, hepatitis, renal insufficiency, splenomegaly, pancytopenia, osteomyelitis, cerebral vasculitis and encephalitis in addition to chorioretinitis [5,13,14]. Patients, after acquiring systemic *M. chimaera* intraoperatively via heater cooler unit bio aerosolization and subsequent colonization of the prosthetic material [15] seems to have a preference for choroidal involvement as an extra-thoracic manifestation [6]. The high oxygen tension of the choroid might be an explanation for the choroidal predilection, since *M. chimaera* is an obligate aerobic bacterium, usually found in tissue with high oxygen levels [6].

The most common ocular findings in the patients with disseminated *M. chimaera* infection are chorioretinal lesions; additional findings include mild anterior and intermediate uveitis, optic disc swelling, and secondary complications including macular and retinal neovascularization [8]. Additional cases of ocular findings in the patients with disseminated *M. chimaera* were recently reported in the United States [16] and Canada [17]. Due to its slow growing rate, symptoms of the infection develop on average 15–17 months post-surgery with an incubation period from 6 weeks to more than 5 years [7]. Mycobacterial infection-related morbidity and mortality is high (greater than 50%), and ophthalmologists can play a key role in reducing this rate by earlier diagnosis, monitoring treatment response and detecting relapses [8,10,16]. A delayed diagnosis might favor the development of extracardiac manifestations and also reduce the response to the treatment, leading to a negative outcome [18].

A common misdiagnosis of *M. chimaera* disseminated infection is systemic sarcoidosis due to its granulomatous inflammation. This was observed in the first patient of our Swiss cohort (case 2) where after an eye examination the patient was initially diagnosed with extrapulmonary sarcoidosis and treated with oral steroids [3,7,8,19]. Ocular involvement in patients with sarcoidosis is common and has been previously reported in 15–25% of the cases [20]. The ocular signs seen in sarcoidosis, if choroidal involvement is present, are not specific for sarcoidosis and can also be seen in other granulomatous involvement such as ocular tuberculosis [21]. Posterior segment involvement occurs in up to one-third of the patients with ocular sarcoidosis, anterior uveitis, vitreous inflammation and cystoid macular edema, and usually leads to blurry vision. However, in our case 2, later on an *M. chimaera* infection was detected in a culture from bone marrow, heparin blood, urine and sputum, and he was treated with clarithromycin, ethambutol and rifabutin according to standard protocol [7]. Despite tuberculostatic therapy, progression of the number and size of choroidal lesions was observed. The ocular lesions reflected systemic disease activity, and ultimately this patient died of splenic rupture and disseminated mycobacterial infection [21].

Differentiating between active and inactive ocular disease by multimodal imaging is important for evaluation of the treatment efficacy, diagnosis of reactivation and breakthrough, and it also helps in guiding these patients [17]. Active lesions appear white/yellowish with indistinct borders on FP represented, whereas inactive or quiet lesions appear as punched-out areas of chorioretinal atrophy with well-defined borders [10]. Hypofluorescent areas observed in the ICGA usually reflect the choriocapillaris hypo- or non-perfusion of the active lesions or atrophic areas of the previous episodes. However, in patients with disseminated *M. chimaera* infection, many lesions regressed completely under treatment, without leading to atrophy or pigmentation. In comparison to lesions observed in non-infectious idiopathic multifocal choroiditis (MFC) and infectious choroiditis including tuberculosis, the lesions observed in disseminated *M. chimaera* usually do not demonstrate pigmentation except for one case in our case series (case 11, Figure 1) [15]. Pigmentation is a part of the healing process and can be observed in different retinal and choroidal infectious and inflammatory diseases. Usually, as the inflammation resolves in these patients the lesions become atrophic with a variable amount of pigment (“punched out” appearance). One explanation might be that in patients with disseminated *M. chimaera* disease, the major part of inflammation is in the choroid and not at the level of the retinal pigmental epithelium (please see Figure 1).

While choroidal lesions found in the tuberculous mycobacterial infections typically resolve with the systemic treatment after a few weeks [22], the nontuberculous mycobacterial infections are very difficult to eradicate and usually require long-term antimicrobial therapy and potential removal of biomaterials. The patients of our cohort were treated using a combination therapy with clarithromycin, ethambutol, rifabutin or rifampin and/or amikacin. In post-cardiovascular surgical *M. chimaera* infections, many clinicians also added companion drugs such as clofazimine or linezolid to the regimen. While linezolid reaches cerebrospinal fluid concentrations of more than 80%, clofazimine does not cross the blood-brain barrier but has shown synergistic effects with amikacin [7,23].

One case (case 9, Figure 1) in this case series demonstrated the critical role of repeated serial ophthalmic examinations with multimodal imaging, which are straightforward, non-invasive and represent an easily accessible indicator of the systemic disease activity. In this case, the progression of the choroidal lesions preceded positive blood culture, which resulted in signaling for early recurrence. Frequent monitoring is therefore recommended in patients where treatment needs to be adapted and after cessation of the treatment. Based on our observations that new ocular lesions developed within 4 to 8 weeks, monthly visits are recommended in patients with discontinuation or adaptation of the systemic treatment [7]. FA and especially ICGA revealed the total lesion number and are better suited to detect overall disease progression than clinical examination, FP and OCT (Figure 1).

The main limitations of this study were its retrospective nature and the limited sample size, which could be explained by the low incidences of this infection. However, to the best of our knowledge, the current study still provides one of the largest datasets and longest follow-up of patients with disseminated *M. chimaera* infections. Ophthalmologists should be aware of this recently described entity and its systemic and ocular findings because they could play a critical role for diagnosis, for monitoring patients under treatment and for evaluation in the follow-up after discontinuation of treatment or treatment adjustment due to adverse drug effect.

## Figures and Tables

**Figure 1 jcm-10-04178-f001:**
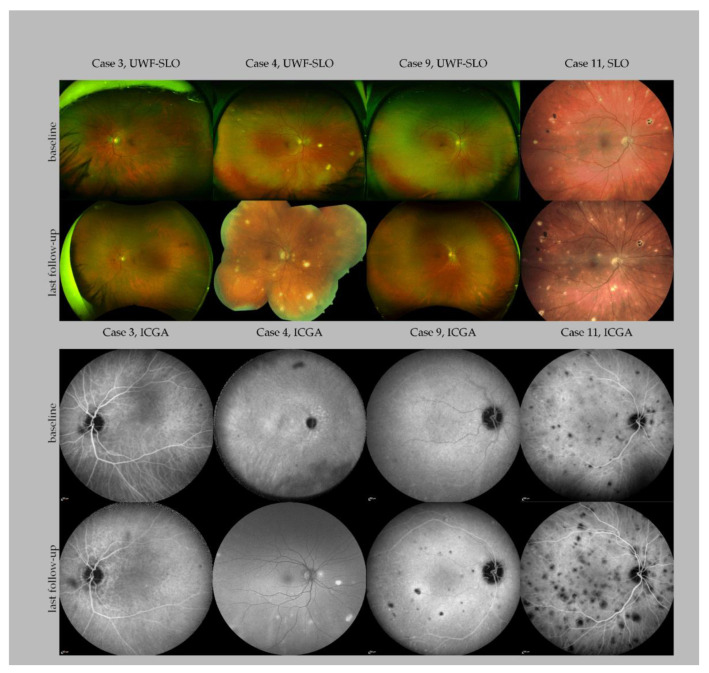
Multimodal imaging of four patients with *Mycobacterium chimaera* infection. The panel showcases baseline and last follow-up imaging of four patients with different disease courses. Case 3 presented without extensive chorioretinal lesions at baseline and never developed more than solitary lesions. Case 4 presented with multiple lesions at baseline and developed widespread chorioretinal lesions at his last follow-up, 36 months after baseline, shortly before his death. A follow-up of ICG could not be performed; a fundus autofluorescence image is shown instead. Case 9 presented with solitary lesions at baseline and developed extensive chorioretinal lesions 47 months after follow-up. Case 11 showed a further progression of the chorioretinal disease over the follow-up period of four months. Note that some of the lesions are pigmented. Abbreviations: UWF: Ultra widefield; SLO: scanning laser ophthalmoscopy; ICGA: indocyanine green angiography.

**Figure 2 jcm-10-04178-f002:**
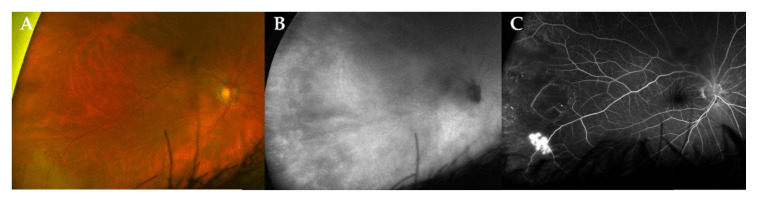
Multimodal imaging of a peripheral retinal neovascularization in a patient with stable disease. Ultra-widefield (UWF) fundus photography (CF, panel (**A**)), indocyanine green angiography fundus imaging (ICG, panel (**B**)) and fluoresceine angiography (FA, panel (**C**)) of a 72-year-old male patient (case 3) at 86 months of follow-up. Note the active retinal neovascularization in the temporal periphery. In this case, the use of ICG could not demonstrate a choroidal lesion.

**Figure 3 jcm-10-04178-f003:**
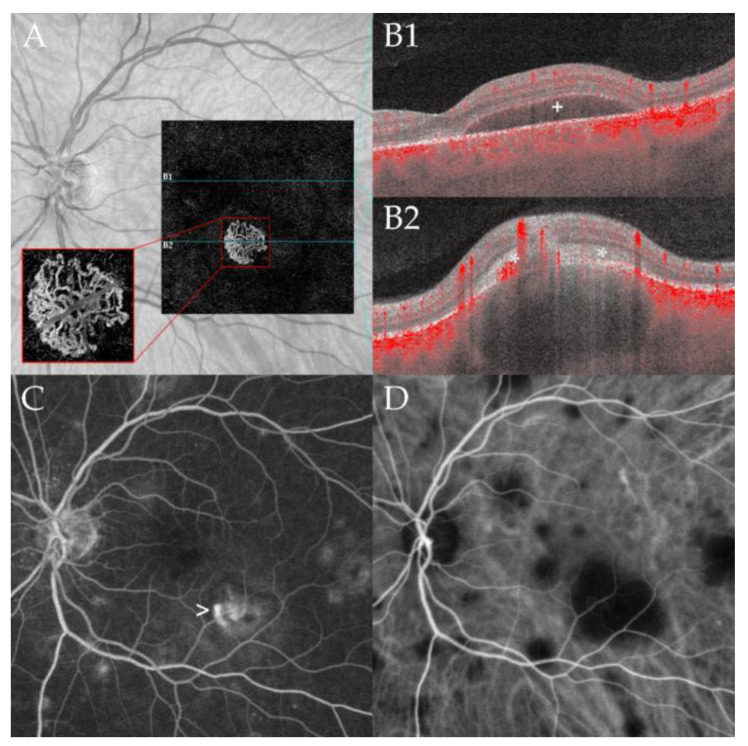
Macular neovascularization associated with choroidal granuloma. Optical coherence tomography angiography (OCTA, panel (**A**)) of a 53-year-old male patient (case 4) at 36 months of follow-up demonstrating a macular neovascular vessels Type 1 in the en face OCTA scan of the outer retina choriocapillaris slab (red rectangle). Note the branching of many small capillaries and the presence of peripheral arcades in the higher magnification of the neovascular membrane. The corresponding B-scan with superimposed flow information (panel (**B1**,**B2**)) demonstrates subretinal fluid (**B1**) and subretinal hyperreflective material (*, (**B2**)) overlying a large choroidal granuloma. Note the subretinal hyperreflective material (*, (**B2**)) and the associated subretinal fluid (+, (**B1**)) leading to decreased vision. Multiple hyperfluorescent lesions at the posterior pole and middle periphery are visible in fluoresceine angiography (FA, panel (**C**)). Note the leakage of the lesion at the posterior pole related to the active neovascularization (arrow). Indocyanine green angiography (ICGA, panel (**D**)) shows disseminated hypofluorescent choroidal lesions including the large granuloma associated with the CNV; the lesions are more numerous compared to FA.

**Figure 4 jcm-10-04178-f004:**
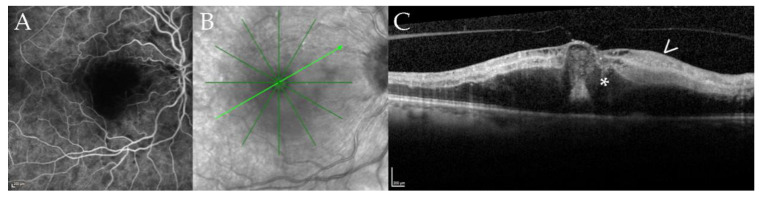
Cilioretinal vessel occlusion in a 67-year-old patient with disseminated *Mycobacterium chimaera* infection. Fluoresceine angiography (FA, panel (**A**)), near infrared imaging (NIR, panel (**B**)), optical coherence tomography B-scan (OCT, panel (**C**)) of case 12 at 18 months of follow-up. Panel (**A**) shows a filling defect of the cilioretinal vessel in the early frames of the FA; the inferotemporal and superotemporal arteries are already filled. Panel (**B**) shows the NIR image with the location of the B-scan (light green arrow) shown in panel (**C**). Note the defocusing of the macular region. The corresponding B-scan (Panel (**C**)) demonstrates hyperreflectivity of the inner retinal layers (arrow) and macular edema (asterisk) secondary due to ischemia after cilioretinal vessel occlusion.

**Table 1 jcm-10-04178-t001:** Patient characteristics at baseline and during the observation period.

Case	Gender	Age at Baseline (Years)	Follow-Up Duration	Activity at Baseline	Reactivation ^2^	Breakthrough ^3^	Deceased	Other Complications
2	Male	51	6 days	Multifocalchoroiditis	No	Yes	Yes	
3	Male	64	88 months	Quiescent ^1^	No	No	No	Retinal neovascularizarion
4	Male	48	41 months	Multifocalchoroiditis	Yes	Yes	Yes	Choroidal neovascularization
5	Male	61	7 months	Multifocalchoroiditis	No	Yes	Yes	
6	Male	62	12 months	Multifocalchoroiditis	No	Yes	Yes	
7	Male	64	48 months	Quiescent ^1^	No	No	No	
8	Male	66	40 months	Quiescent ^1^	No	Yes	No	
9	Male	50	47 months	Quiescent ^1^	Yes	No	No	
11	Female	55	4 months	Multifocalchoroiditis	No	No	No	
12	Male	66	18 months	Quiescent ^1^	No	No	No	Cilioretinal vessel occlusion

^1^ Quiescent refers to mild ocular abnormalities (only very few inactive (2–5/eye) choroidal lesions). ^2^ Reactivation refers to recurrence of disease activity after adaptation/stopping of therapy. ^3^ Breakthrough refers to recurrence of disease activity under continuous therapy.

## Data Availability

Data will be made available upon reasonable request to the corresponding author.

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
