# Peer review of "Long-Term Clinical and Multimodal Imaging Findings in Patients with Disseminated Mycobacterium Chimaera Infection"

_jcm, 2021, doi:10.3390/jcm10184178_

Round 1
Reviewer 1 Report
I appreciate the work of Zweifel and colleagues, it’s well written and highlights the importance of the ophthalmologist consultation during M. Chimaera disseminated infection.
To the authors: please highlight in the discussion section the importance of anti-mycobacterial drug that can pass the B-B- barrier (such as linezolid and bedaquiline, while clofazimine does not pass it. Please cite doi: 10.2217/fmb-2019-0231.)
Author Response
Dear Reviewer,
We are grateful to you for your time and constructive comments on our manuscript.
We have amended the manuscript according to your comments and valuable suggestions. Changes in the last version of the manuscript are reported as red tracked changes.
Below, we also provide a point-by-point response explaining how we have addressed each of your comments.
POINT-BY-POINT RESPONSE
English language and style
() Extensive editing of English language and style required
(x) Moderate English changes required
( ) English language and style are fine/minor spell check required
( ) I don't feel qualified to judge about the English language and style
AUTHORS’ REPLY:
Thank you for the valuable comments. According your suggestion, our paper has undergone professional English editing by Resapro.
REVIEWER 1:
Comments to Authors
I appreciate the work of Zweifel and colleagues, it’s well written and highlights the importance of the ophthalmologist consultation during M. Chimaera disseminated infection.
To the authors: please highlight in the discussion section the importance of anti-mycobacterial drug that can pass the B-B- barrier (such as linezolid and bedaquiline, while clofazimine does not pass it. Please cite doi: 10.2217/fmb-2019-0231.)
AUTHORS’ REPLY:
We thank the reviewer for the valuable comment about the manuscript and the antimicrobial therapy. We adapted the corresponding section in the manuscript and now cite the reference as suggested by the reviewer (page 9, line 316-321).
We hope to receive your favorable consideration for our paper
Best regards,
All coauthors

Reviewer 2 Report
The authors present longer term follow-up of ten previously published patients. In this series, they track the clinical outcomes of these patients, as well as their ophthalmologic findings. Readers are referred to previous publications featuring these patients for additional clinical details, and the case numbering has been maintained between publications to allow for cross-reference.
From a general medicine perspective, the most important conclusion is the correlation of active chorioretinal lesions with on-going disease activity - i.e.: reactivation, breakthrough, and, ultimately, mortality. This is well demonstrated in Table 1. The case selected for more detailed dissection highlight how serial ophthalmologic examination can identify changes that correlate with clinical status and can help guide treatment decisions, including the decision to resume or intensify multi-agent antimicrobial therapy, and even to anticipate a possible need for re-redo cardiac surgery. The advantages of ophthalmologic examination are explored further in the discussion - It is readily accessible, and offers a more rapid objective and quantifiable assessment of the status of the underlying M. chimaera than cultures, which may take additional weeks to ring positive. This reports builds a strong case supporting the routine engagement of ophthalmologists for frequent serial examination in M. chimaera patients.
I do wonder whether there is any correlation between the ophthalmologic findings and neuro-cognitive findings. The presumptive mechanism of choroidal seeding is hematologic spread, but due to proximity and continuity of the eye with the central nervous system, did the authors notice any neurologic implications of specific findings?
Author Response
Dear Reviewer,
We are grateful to you for your time and constructive comments on our manuscript.
We have amended the manuscript according to your comments and valuable suggestions. Changes in the last version of the manuscript are reported as red tracked changes.
Below, we also provide a point-by-point response explaining how we have addressed each of your comments.
POINT-BY-POINT RESPONSE
English language and style
( ) Extensive editing of English language and style required
( ) Moderate English changes required
(x) English language and style are fine/minor spell check required
( ) I don't feel qualified to judge about the English language and style
AUTHORS’ REPLY:
Thank you for the valuable comments. According your suggestion, our paper has undergone professional English editing by Resapro.
REVIEWER 2
Comments to Authors
The authors present longer term follow-up of ten previously published patients. In this series, they track the clinical outcomes of these patients, as well as their ophthalmologic findings. Readers are referred to previous publications featuring these patients for additional clinical details, and the case numbering has been maintained between publications to allow for cross-reference.
From a general medicine perspective, the most important conclusion is the correlation of active chorioretinal lesions with on-going disease activity - i.e.: reactivation, breakthrough, and, ultimately, mortality. This is well demonstrated in Table 1. The case selected for more detailed dissection highlight how serial ophthalmologic examination can identify changes that correlate with clinical status and can help guide treatment decisions, including the decision to resume or intensify multi-agent antimicrobial therapy, and even to anticipate a possible need for re-redo cardiac surgery. The advantages of ophthalmologic examination are explored further in the discussion - It is readily accessible, and offers a more rapid objective and quantifiable assessment of the status of the underlying M. chimaera than cultures, which may take additional weeks to ring positive. This reports builds a strong case supporting the routine engagement of ophthalmologists for frequent serial examination in M. chimaera patients.
I do wonder whether there is any correlation between the ophthalmologic findings and neuro-cognitive findings. The presumptive mechanism of choroidal seeding is hematologic spread, but due to proximity and continuity of the eye with the central nervous system, did the authors notice any neurologic implications of specific findings?
AUTHORS’ REPLY:
We would like to thank the reviewer for raising this important aspect. We agree that some patients do develop neurological complications including vasculitis of the brain and encephalitis in addition to chorioretinitis. We observed one patient (case 6), who developed a rapid neurocognitive decline during the course of the disease. He did not receive an MRI due to his rapid clinical decline, but the electroencephalogram shortly before death showed discontinuous rhythmic severe focal left hemispheric findings without epilepsy-typical potentials.
We adapted the manuscript and now mention neurologic sequelae. We also added two additional references covering this issue. In one paper by Lau et al. (ref. 14) histopathologic evidence of brain granulomas is described as well. Please see the adapted discussion section of the manuscript (page 8, lines 258-261).
“This is due to the disseminated granulomatous inflammatory manifestations including endocarditis, vascular graft infection, bacteremia, hepatitis, renal insufficiency, splenomegaly, pancytopenia, osteomyelitis, cerebral vasculitis and encephalitis in addition to chorioretinitis.”
New added references:
- Tan N, Sampath R, Abu Saleh OM, Tweet MS, Jevremovic D, Alniemi S, Wengenack NL, Sampathkumar P, Badley AD. Disseminated Mycobacterium chimaera Infection After Cardiothoracic Surgery. Open Forum Infect Dis. 2016 Jun 16;3(3):ofw131. doi: 10.1093/ofid/ofw131. PMID: 27703994; PMCID: PMC5047393.
- Lau D, Cooper R, Chen J, Sim VL, McCombe JA, Tyrrell GJ, Bhargava R, Adam B, Chapman E, Croxen MA, Garady C, Antonation K, van Landeghem FKH, Ip S, Saxinger L. Mycobacterium chimaera Encephalitis Following Cardiac Surgery: A New Syndrome. Clin Infect Dis. 2020 Feb 3;70(4):692-695. doi: 10.1093/cid/ciz497. PMID: 31247065.
We hope to receive your favorable consideration for our paper
Best regards,
All coauthors
